# PCDVQ: Enhancing Vector Quantization for Large Language Models via Polar Coordinate Decoupling

## Abstract

Large Language Models (LLMs) face significant challenges in edge deployment due to their massive parameter scale. Vector Quantization (VQ), a clustering-based quantization method, serves as a prevalent solution to this issue for its extremely low-bit (even at 2-bit) and considerable accuracy. Since a vector is a quantity in mathematics and physics that has both direction and magnitude, existing VQ works typically quantize them in a coupled manner. However, we find that direction exhibits significantly greater sensitivity to quantization compared to the magnitude. For instance, when separately clustering the directions and magnitudes of weight vectors in LLaMA-2-7B, the accuracy drop of zero-shot tasks are 46.5% and 2.3%, respectively. This gap even increases with the reduction of clustering centers. Further, Euclidean distance, a common metric to access vector similarities in current VQ works, places greater emphasis on reducing the magnitude error. This property is contrary to the above finding, unavoidably leading to larger quantization errors. To these ends, this paper proposes Polar Coordinate Decoupled Vector Quantization (PCDVQ), an effective and efficient VQ framework consisting of two key modules: 1) Polar Coordinate Decoupling (PCD), which transforms vectors into their polar coordinate representations and perform independent quantization of the direction and magnitude parameters. 2) Distribution Aligned Codebook Construction (DACC), which optimizes the direction and magnitude codebooks in accordance with the source distribution. Experimental results show that PCDVQ outperforms baseline methods at 2-bit level by at least 1.5% zero-shot accuracy, establishing a novel paradigm for accurate and highly compressed LLMs.

## 1 Introduction

Large language models (LLMs) like GPT Brown et al. (2020); Ouyang et al. (2022), LLaMA Touvron et al. (2023a;b); Grattafiori et al. (2024), and DeepSeek Liu et al. (2024a); Guo et al. (2025) play a crucial role in natural language processing, exhibiting extraordinary capabilities in understanding and generating text. However, their large size poses significant challenges for deployment. Especially, the large number of weights in LLMs consumes a considerable amount of memory. For instance, the LLaMA-3-70B model needs approximately 140GB of memory when stored in FP16 format, which exceeds the capabilities of high-end GPUs and requires multi-GPU deployment.

Post-training Quantization (PTQ) is an essential technique for reducing the memory occupy of weights, thereby promoting the edge deployment of models. Scaler Quantization (SQ) Frantar et al. (2022); Lin et al. (2023); Xiao et al. (2022); Shao et al. (2023); Yao et al. (2022), which converts each scalar weight in the model into lower bit-width, is the most common PTQ approach for its stable quantization ability in the 3~4 bit setting. However, owing to the constrains of numerical representation, SQ struggles to achieve extremely low-bit levels. On the other hand, Vector Quantization (VQ) Chee et al. (2023); Tseng et al. (2024a); Van Baalen et al. (2024); Liu et al. (2024b); Tseng et al. (2024b) exhibits superior performance compared to SQ at 2~3 bits, thus gaining more research attention recently. It typically regards the original model weights as high-dimensional vectors from the origin (where all dimensions is zero) to their values, then represents all of them with a finite subset (i.e., codebook).

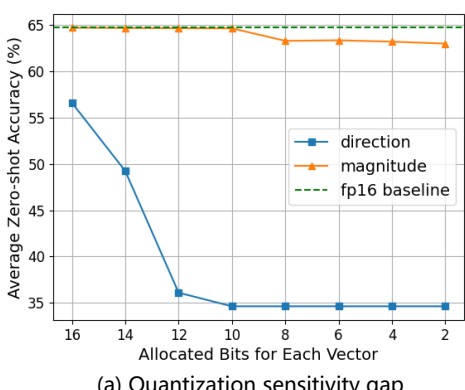
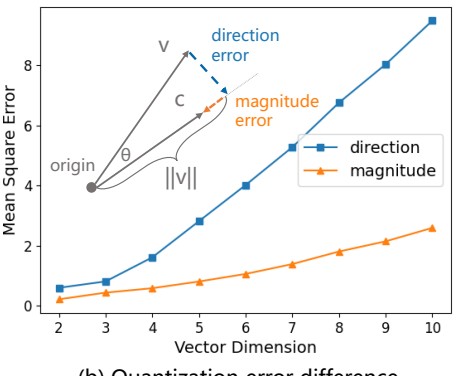

(a) Quantization sensitivity gap

(b) Quantization error difference

Figure 1: Comparison between direction and magnitude. We utilize the classic K-Means Liu et al. (2024b) algorithm to perform VQ, which is also employed in VPTQ Liu et al. (2024b). The LLaMA-2-7B model is utilized for representation. In sub-figure (a), we separately quantize the direction and magnitude of all weight vectors. The x-axis denotes the bits of the index (i.e., $2^x$ of clusters), and the y-axis represents the average accuracy of five zero-shot tasks. In sub-figure (b), we directly perform VQ. Given a vector $v$ and its quantized center $c$, the direction error is calculated by $2\|v\|^2(1-\cos\theta)$, where $\theta$ is the angle between $v$ and $c$. The magnitude error is calculated by $(\|v\| - \|c\|)^2$. The x-axis is the dimension setting of VQ, and the y-axis is MSE.

Despite the promising development of VQ techniques, we have noticed a crucial issue that hinder their accuracy improvement. Given that a vector inherently comprises both direction and magnitude, existing VQ works typically quantize them in a coupled manner. However, we find that **direction exhibits greater sensitivity to quantization compared to its magnitude counterpart.** For instance, when separately clustering the directions and magnitudes of weight vectors in LLaMA-2-7B Touvron et al. (2023b), the accuracy drop of the direction quantization is much more significant with the reduction of bit-width. As shown in Figure 1 (a), direction quantization can decrease about 30% of the fp16 performance, while that for magnitude quantization is solely 3%. This gap can be caused by the dimensional difference. The direction obtains a larger spatial degree of freedom than the magnitude, thus requiring more cluster centers for representation.

More even, Euclidean distance, a common metric in current VQ methods for accessing the vector similarity, **emphasizes reducing the magnitude error**. This property is contrary to the above finding and can unavoidably lead to larger quantization errors. For instance, we evaluate direction and magnitude error under the same unit with Mean Square Error (MSE) as described in Figure 1 (b). Results show that the magnitude error is always smaller and increases slower with the vector dimension than that of the direction. Our analysis reveals that the magnitude error contributes quadratically to MSE, whereas the direction error exerts an approximately linear effect.

To these ends, we introduce PCDVQ, an effective VQ framework tailored for LLMs. (1) We introduces a polar coordinate decoupling (PCD) technique. It converts input vectors into their polar coordinate representations and performs independent quantization on both direction and magnitude parameters. Subsequently, longer bit-width is allocated to the direction codebook and the cosine similarity can be utilized for evaluating the direction distance. (2) We introduce a distribution aligned codebook construction (DACC) method to establish two distinct codebooks for direction and magnitude parameters in accordance with their distribution. For uniformly distributed directions we utilize the greedy algorithm to sample directions of the $E_8$ lattice Viazovska (2017). As for the magnitude, it follows the root distribution of $k$-dimensional Chi-square distribution ($\mathcal{X}^2(k)$), where $k$ is the vector dimension. By deriving the expression of its Probability Density Function (PDF), the Lloyd-Max algorithm Lloyd (1982) can be applied to achieve the optimal scalar quantization.

Experiments show that PCDVQ outperforms existing weight-only PTQ methods across various tasks and LLMs, including LLaMA-2 Touvron et al. (2023b), LLaMA-3 Grattafiori et al. (2024), and Mistral-7B Jiang et al. (2023). Overall, PCDVQ achieves averagely 1.5% accuracy improvement

compared to the state-of-the-art (SOTA) baseline methods at 2-bit level on five typical zero-shot tasks across all models. It can also significantly decrease the perplexity (PPL) on both WikiText2 Merity et al. (2016) and C4 Raffel et al. (2020) datasets. Lastly, our contributions can be concluded as follows:

- We identify that the direction and magnitude present different sensitivities to VQ, while the former one typically encounters much larger quantization losses. Our analysis further reveals that existing approaches of codebook construction and similarity measurement are inadequate for addressing this issue.

- We propose PCDVQ. Firstly, it introduces the polar coordinate decoupling to separately quantize directions and magnitudes, while allocating more bits to the direction codebook. Secondly, it introduces two distribution-aligned codebook for the direction and magnitude.

- We evaluate PCDVQ on both PPL and zero-shot tasks. Experiments show that PCDVQ can effectively address the problem of different sensitivities of direction and magnitude to quantization, further promoting the development of accurate and highly-compressed LLMs.

## 2 PRELIMINARIES

Post Training Quantization (PTQ) Yuan (2024); Yue et al. (2024) is an essential technique for deploying neural networks in environments with limited computational resources. For LLMs, their vast sizes of parameters account for the major memory consumption and bandwidth burden during inference. Thereby, this paper focus on the weight-only PTQ. Current methods can be categorized into two types, including scalar quantization (SQ) and vector quantization (VQ).

Notably, recent works Wu et al. (2025) and Han et al. (2025) also propose to leverage the polar coordinate in quantization. These are SQ methods designed for the KV cache quantization. Differently, our work focuses on enhancing the weight-only VQ. Moreover, we introduce polar coordinates to decouple the direction and magnitude of vectors defined in Cartesian coordinates, which represents an innovative motivation.

### 2.1 SCALAR QUANTIZATION

Weight-only SQ converts weights of pretrained neural networks from high precision (e.g., 16-bit floating point numbers) to lower precision (e.g., 4-bit integers). Given a weight $\mathbf{W}$, it is typically implemented with symmetrical- and uniform- quantization as:

$$\mathrm{SQ}(\boldsymbol{W}) = \mathrm{clamp}(\lfloor \frac{\boldsymbol{W}}{\boldsymbol{s}} \rceil, -2^{b-1}, 2^{b-1} - 1), \quad \boldsymbol{s} = \frac{\max(|\boldsymbol{W}|)}{2^{b-1} - 1}, \tag{1}$$

where $\mathrm{SQ}(\cdot)$ is the SQ operation, $s$ is the scale factor, $\lfloor \cdot \rceil$ denotes the rounding-to-nearest operator, $b$ is the quantization bit-width, and $\mathrm{clamp}$ is the clipping function.

Based on this progress, several optimization methods have been proposed. For instance, GPTQ Frantar et al. (2022) introduces a layer-wise quantization technique based on approximate second-order information. AWQ Lin et al. (2023) protects 1% pivotal weights not to be quantized by introducing an activation-aware smoothing operation. QuaRot Ashkboos et al. (2024) applies the Hadamard matrix for rotation which significantly suppresses the outliers. Despite their improvement of accuracy, such scalar-based methods are limited to 3∼4 bit level. When it comes to a more aggressive bit-width, these methods can suffer from substantial performance degradation.

### 2.2 VECTOR QUANTIZATION

Within 2-bit level weight-only PTQ, VQ has gained more research attention for its advantage in modeling the raw data distribution Tseng et al. (2024a) and high compression ratio. Given a weight $\boldsymbol{W}$ with $p$ rows and $q$ columns to be quantized, VQ reshapes it into $\boldsymbol{W}'$ with dimensions $(p * q/k, k)$. For each $k$-dimensional row vector, VQ replaces it with the $n$-bit index of the nearest vector from the codebook $\boldsymbol{C} \in \mathbb{R}^{2^n \times k}$. Typically, the Euclidean distance (calculated by the Frobenius normalization $|| \cdot ||_F$) is taken to measure similarities. In this case, the quantization process can be expressed as:

$$\mathrm{VQ}(\boldsymbol{W}') = \{\underset{j \in 2^n}{\mathrm{argmin}} ||\boldsymbol{W}'_{i,:} - \boldsymbol{C}_{j,:}||_F \,|\, i = 1, ..., p * q/k\}. \tag{2}$$

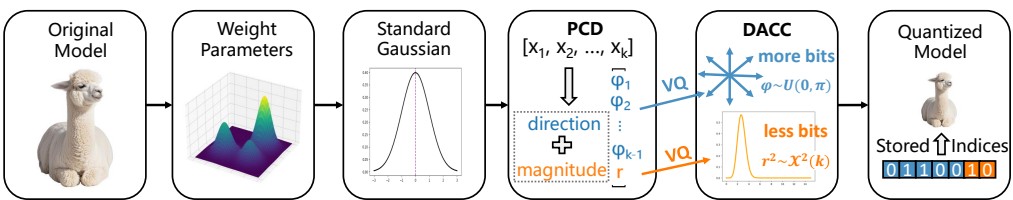

Figure 2: Pipeline of PCDVQ. It contains two novel techniques.

The codebook $C$ has shape $(2^k, d)$, where each row vector represents a cluster center. Current VQ methods propose different strategies to optimize the codebook construction. For instance, VPTQ Liu et al. (2024b) and GPTVQ Van Baalen et al. (2024) cluster the source vectors by K-Means Algorithm Lloyd (1982) and Expectation-Maximization Algorithm Moon (1996), respectively. AQLM Egiazarian et al. (2024) further utilizes layer-wise training for the codebook to obtain higher accuracy. QuIP# Tseng et al. (2024a) applies the Hadamard rotation to suppress outliers and introduces a pre-defined global codebook named E8P.

## 3 METHODS

In this section, we analyze the different sensitivities of direction and magnitude to VQ, emphasizing the importance of direction quantization. Next, we introduce the PCDVQ framework. Given a model to be quantized, PCDVQ quantizes its weights one by one. For a weight parameter of a linear layer, it is first regularized into the standard Gaussian distribution (Section 3.2.1). Secondly, we apply PCD (Section 3.2.2). It transforms vectors into polar coordinates, then independently quantizing the direction and magnitude parameters with DACC (Section 3.2.3), which establishes codebooks aligned with the standard Gaussian distribution. The pipeline of PCDVQ is shown in Figure 2.

### 3.1 MOTIVATION

#### 3.1.1 HIGHER QUANTIZATION SENSITIVITY IN DIRECTION

As mentioned in Section 1, direction is identified to be more essential than magnitude for its higher quantization sensitivity to VQ. This phenomenon stems from the dimensional discrepancy. Data points in high-dimensional spaces tend to exhibit higher sparsity Hastie et al. (2009); Bishop & Nasrabadi (2006); Donoho et al. (2000), thereby diminishing the representational capacity of clusters. Given any $k$-dimensional vector $v \in \mathbb{R}^k$, it can be decomposed to its direction component $d$ and magnitude component $m$:

$$v = \underbrace{\frac{v}{||v||}}_{d} \cdot \underbrace{||v||}_{m},$$ (3)

where $|| \cdot ||$ denotes the magnitude computation. While direction component $d$ is $k$-dimensional, the magnitude $m$ remains single-dimensional. Given any of their cluster centers ($c_d \in \mathbb{R}^k$ and $c_m \in \mathbb{R}^1$), the typical calculation of similarities by the Euclidean distance $D$ can be expressed as:

$$D(d, c_d) = \sqrt{\sum_{i=0}^{i=k}(d_i - c_{d_i})^2}, \qquad D(m, c_m) = \sqrt{(m_i - c_{m_i})^2}.$$ (4)

While the direction distance $D(d, c_d)$ obtains more terms for its high dimension, it is more potential to achieve a larger value than the magnitude distance $D(m, c_m)$. Thereby, allocating more representative abilities of codebooks to the direction appears to be crucial for VQ frameworks.

#### 3.1.2 INADEQUACY OF EUCLIDEAN DISTANCE

While previous studies have demonstrated that VQ requires higher vector dimension to enhance accuracy Tseng et al. (2024a;b), the sensitivity discrepancy between direction and magnitude can

become correspondingly more significant. However, the Euclidean distance metric is more sensitive to magnitude quantization errors than to direction discrepancies. Given any vector $\boldsymbol{v}$ and its quantized version $\hat{\boldsymbol{v}}$, the typical MSE quantization loss $\mathcal{L}$ can be calculated by:

$$\mathcal{L}(\boldsymbol{v}, \hat{\boldsymbol{v}}) = D(\boldsymbol{v}, \hat{\boldsymbol{v}})^2 = (\Delta r)^2 + 2 \cdot ||\boldsymbol{v}|| \cdot ||\hat{\boldsymbol{v}}|| \cdot (1 - \cos \Delta \theta),$$
$$\text{where} \quad \Delta r = \sqrt{(||\boldsymbol{v}|| - ||\hat{\boldsymbol{v}}||)^2}, \quad \Delta \theta = \frac{\boldsymbol{v} \cdot \hat{\boldsymbol{v}}}{||\boldsymbol{v}|| \cdot ||\hat{\boldsymbol{v}}||}. \tag{5}$$

It can be observed that the magnitude gap $\Delta r$ contributes quadratically to Eq. 5, whereas the direction gap $\Delta \theta$ exerts an approximately linear effect.

### 3.2 POLAR COORDINATE DECOUPLED VECTOR QUANTIZATION

#### 3.2.1 STANDARD GAUSSIAN REGULARIZATION

Previous SQ methods Chee et al. (2023); Ashkboos et al. (2024); Liu et al. (2024c); Hu et al. (2025) and VQ methods Tseng et al. (2024a) primarily employ the randomized Hadamard matrix (RHM) to suppress outliers. In contrast, our work leverages this transformation to convert model parameters into a standard Gaussian distribution ($\mathcal{N}(0, 1)$).

Given a weight parameter $\boldsymbol{w} \in \mathbb{R}^{p \times q}$, the transformation is performed column by column. For each column vector $\boldsymbol{x} \in \mathbb{R}^{p \times 1}$ if $\boldsymbol{S} \in \mathbb{R}^{p \times p}$ is a RHM then $\boldsymbol{S} \cdot \boldsymbol{x}$ approximately follows the Gaussian distribution ($\mathcal{N}(0, \frac{||\boldsymbol{x}||^2}{p})$) Chee et al. (2023). Subsequently, the standard Gaussian transformation is applied with the scaling factor $s = \frac{||\boldsymbol{x}||}{\sqrt{p}}$. Since this factor is shared by all positions of a column, the resulting storage overhead can be neglected.

This application achieves two key advantages: (1) it integrates the distributions across diverse model weights, and (2) it enables mathematical formulation of these distributions. Thereby, the construction of codebooks is provided with a unified formulation for reference, facilitating their alignment to the distribution to be quantized.

#### 3.2.2 POLAR COORDINATE DECOUPLING

Current VQ codebooks store vector features, making it inflexible to adjust their representative abilities for direction and magnitude. To this end, we propose PCD, which explicitly extracts the direction and magnitude parameters by transforming the cartesian coordinates of vectors to polar coordinates. Specifically, given a vector parameter $\boldsymbol{v} = \{\boldsymbol{v}_i \,|\, i = 1, 2, ..., k\}$ of a model weight to be quantized, it can be transformed to $\boldsymbol{v}' = \{\phi_1, \phi_2, ..., \phi_{k-1}, r\}$ by:

$$\phi_i = \text{atan2}(\sqrt{\sum_{j=i+1}^{j=k} \boldsymbol{v}_j^2}, \boldsymbol{v}_i), i = 1, 2, ..., k-1, \quad r = \sqrt{\sum_{j=1}^{j=k} \boldsymbol{v}_j}, \tag{6}$$

where $\text{atan2}$ is the two-argument arc-tangent function, $r$ is the single-dimensional magnitude parameter, and $\phi$ denotes the direction parameter ($\phi_{k-1} \in [0, 2\pi]$, while others $\in [0, \pi]$).

Subsequently, the direction and magnitude parameters can be independently quantized by two distinct codebooks $\boldsymbol{C}_\phi \in \mathbb{R}^{2^a \times (k-1)}$ and $\boldsymbol{C}_r \in \mathbb{R}^{2^b \times 1}$:

$$\text{VQ}_\phi(\boldsymbol{\phi}) = \underset{j \in 2^a}{\text{argmax}} \, \cos(\boldsymbol{\phi}, \boldsymbol{C}_{\phi_{j,:}}), \quad \text{VQ}_r(r) = \underset{j \in 2^b}{\text{argmin}} \sqrt{(r - \boldsymbol{C}_{r_j})^2}, \tag{7}$$

where $\boldsymbol{\phi} = \{\phi_i \,|\, i = 1, 2, ..., k-1\}$ and $\cos(\cdot)$ denotes the calculation of cosine similarity. The symbols $a$ and $b$ represent the allocated bits for direction and magnitude, respectively. It should be satisfied that the summarization of $a$ and $b$ is equal to the total bits of the vector index within conventional VQ methods. Consequently, the quantized $\hat{\boldsymbol{v}}'$ of the polar coordinate representation $\boldsymbol{v}'$ can be obtained by splicing the respective indices of its direction and magnitude components:

$$\hat{\boldsymbol{v}}' = [\text{VQ}_\phi(\boldsymbol{\phi}), \text{VQ}_r(r)]. \tag{8}$$

Thus, one can assign more representations to direction by setting $a$ to a relatively larger value than $b$. Given more cluster centers, the larger error of direction quantization can be effectively reduced.

### 3.2.3 DISTRIBUTION ALIGNED CODEBOOK CONSTRUCTION

After the standard gaussian regularization, all variables to be quantized follow the standard Gaussian distribution. Notably, PCD does not change this property, and performs VQ on direction parameter $\phi$ and magnitude parameter $r$ in an independent way. Based on this, we propose the distribution aligned codebook construction (DACC) technique. Firstly, we analyze that the direction follows the spatially uniform distribution, while the magnitude follows the root Chi-square distribution. Secondly, we construct the codebooks to maximize their representative abilities.

**Spatially Uniform Direction Codebook** It is a fact that directions of Gaussian variables are uniformly distributed in space. For a direction parameter $\phi$, it should have:

$$\phi_i \sim \begin{cases} U(0, \pi) & \text{if} \quad i \neq k-1 \\ U(0, 2\pi) & \text{if} \quad i = k-1 \end{cases}. \tag{9}$$

In the 8-dimensional space, $E_8$ lattice Viazovska (2017) is proved to achieve the densest packing of spheres, and its directions $\phi_{E_8}$ are highly uniform and symmetric in space. Considering these characteristics, we greedily sample directions from $\phi_{E_8}$ as described in Algorithm 1.

---

**Algorithm 1:** Greedy algorithm for constructing the direction codebook

    **input** : All directions of $E_8$ lattice $\phi_{E_8}$ and allocated bits $a$ for direction
    **output** : Direction codebook $C_\phi$

1   Randomly select one initial direction $\phi_1$ from $\phi_{E_8}$ and set $C_{\phi 1}$ to $\phi_1$;
2   **for** $i \leftarrow 2$ **to** $2^a$ **do**
3      $min\_max\_cos \leftarrow 1e9$;
4      $selected \leftarrow None$;
5      **foreach** *direction* $\phi$ *in* $\phi_{E_8}$ **do**
6         **if** $\phi \in C_\phi$ **then**
7            continue;
8         $current\_max \leftarrow \max_{\phi_c \in C_\phi} \cos(\phi, \phi_c)$;
9         **if** $current\_max < min\_max\_cos$ **then**
10           $min\_max\_cos \leftarrow current\_max$;
11           $selected \leftarrow \phi$;
12      $C_{\phi i} \leftarrow selected$;

---

Consequently, the codebook $C_\phi$ represents a finite set of relatively uniform directions in 8-dimensional space, which is aligned with the direction distribution of Gaussian variables. Notably, this process is offline and performed only once for all circumstances since all transformed weights follow $\mathcal{N}(0, 1)$. We also compare this approach with other common methods in Section 4.3.

**Root Chi-square Magnitude Distribution** The sum of squares of k independent standard Gaussian variables follows a Chi-square distribution with k degrees of freedom, which is typically denoted by $\mathcal{X}^2(k)$. The magnitude $r$ follows the root distribution of $k$-dimensional Chi-square distribution, which can be expressed as:

$$r^2 \sim \mathcal{X}^2(k). \tag{10}$$

Thereby, the probability density function (PDF) $f(r)$ and cumulative distribution function (CDF) $F(r)$ of the magnitude distribution can be derived through this relationship:

$$f(r) = \frac{2^{1-\frac{k}{2}}}{\Gamma(\frac{k}{2})} r^{k-1} e^{-\frac{r^2}{2}}, \quad F(r) = \frac{\gamma(\frac{k}{2}, \frac{r^2}{2})}{\Gamma(\frac{k}{2})}, \quad r \geq 0,$$

$$\text{where} \quad \Gamma(r) = \int_0^\infty t^{r-1} e^{-t} dt, \quad \gamma(r, z) = \int_0^z t^{r-1} e^{-t} dt. \tag{11}$$

Detailed proof are provided in A.2. Since the distribution characteristics of the magnitude are analytically describable as shown in Eq 11, the conditions of the llyod-max algorithm Lloyd (1982) are satisfied. We apply this methods because it is proved to be the optimal non-uniform scalar quantization (i.e., 1-dimensional VQ), which is described in Algorithm 2.

---

**Algorithm 2:** Llyod-max algorithm for constructing the magnitude codebook

**input** : Allocated bits $b$ for magnitude, maximum threshold $\tau$, terminate threshold $tol$, and max iteration $M$

**output** : Magnitude codebook $C_r$

1   Compute $max\_r$ to satisfy $F(max\_r) = \tau$;

2   Uniformly select initial magnitudes $C_r = \{r_i \,|\, i = 1, 2, .., 2^b\}$ from $[0, max\_r]$;

3   **for** $m \leftarrow 1$ **to** $M$ **do**

4      $u_0 \leftarrow 0, \quad u_j \leftarrow \frac{r_i + r_{i+1}}{2}, \quad i = 1, 2, ..., 2^b - 1, \quad u_{2^b} \leftarrow max\_r$;

5      $loss = \{l_i = 0 \,|\, i = 1, 2, ..., 2^b\}$;

6      **for** $i \leftarrow 1$ **to** $2^b$ **do**

7         $cur \leftarrow \frac{\int_{u_{i-1}}^{u_i} t f(t) dt}{F(u_i) - F(u_{i-1})}$;

8         $r_i \leftarrow cur$;

9         $loss_i \leftarrow |cur - r_i|$;

10      **if** $\max l_i \in loss < tol$ **then**

11         break;

12      $C_r \leftarrow \{r_i \,|\, i = 1, 2, ..., 2^b\}$;

---

# 4 EXPERIMENTS

## 4.1 EXPERIMENTAL SETTINGS

**Models & Datasets & Baselines**   We evaluate PCDVQ on LLaMA-2 series Touvron et al. (2023b) (LLaMA-2-7B, LLaMA-2-13B, and LLaMA-2-70B), LLaMA-3 series Grattafiori et al. (2024) (LLaMA-3-8B and LLaMA-3-70), and Mistral-7B Jiang et al. (2023). Following previous PTQ works Shao et al. (2023); Ashkboos et al. (2024); Hu et al. (2025); Liu et al. (2024b), we select WikiText-2 and C4 datasets for the perplexity (PPL) evaluation. We also perform the common QA evaluation on five zero-shot datasets, including Arc-Challenge Clark et al. (2018), Arc-Easy Clark et al. (2018), HellaSwag Zellers et al. (2019), PIQA Bisk et al. (2020), and WinoGrande Sakaguchi et al. (2021). Since this work focuses on the weight quantization for LLMs, we compare PCDVQ with both SQ and VQ methods under 2-bit level weight-only quantization, including GPTQ Frantar et al. (2022), GPTVQ Van Baalen et al. (2024), DB-LLM Chen et al. (2024), QuIP Chee et al. (2023), QuIP# Tseng et al. (2024a), AQLM Egiazarian et al. (2024), and VPTQ Liu et al. (2024b).

**Implementation Detail**   As mentioned in Section 3.2.2, the PCD technique introduces hyper-parameters $a$ and $b$ to control the allocation of VQ representations. While $b$ is fixed to 2, we set $a$ to 14 for 2-bit quantization and 16 for 2.25-bit quantization. Fine-tuning is a common technique for enhancing the performance of VQ Tseng et al. (2024a); Van Baalen et al. (2024); Egiazarian et al. (2024); Liu et al. (2024b), which simply adjusts the un-quantized weights of linear layers or the parameters of normalization layers. Since this work does not focus on the design of this approach, we directly employ the block-wise fine-tuning and end-to-end (e2e) fine-tuning of Tseng et al. (2024a). The parameter configurations of these methods remain consistent with their original implementations, except that we employ randomly selected samples from the training splits of WikiText2 and C4 .

## 4.2 MAIN RESULTS

Our proposed PCDVQ outperforms all baseline methods in the 2-bit level weight-only quantization setting as shown in Tables 1 and 2. Specifically, for baselines less than 2.1-bit, PCDVQ (2-bit) exhibit the best accuracy on all tasks and models. Furthermore, it can also achieve comparable performance to current VQ methods even with a lower bit-width. For instance, its QA Avg results are higher than all baselines (larger than 2-bit) across LLaMA-2-70B, LLaMA-3-70B, and Mistral-7B.

PCDVQ (2.25-bit) introduces one more bit-width to extend the representative ability of the direction codebook. It significantly enhances the overall performance of its 2-bit version, demonstrating the effectiveness of the combination of our proposed PCD and DACC.

Table 1: Quantization performance comparison on LLaMA-2 series. The symbol '-' denotes no quantization is applied. The arrow ↓ means that a lower number represents a better performance (↑ is of the same sense). The context length of the WikiText2 and C4 evaluation is 4096. QA Avg is the average result of the five zero-shot evaluation tasks. Full results are in A.3. The best performance for each task is highlighted in bold, and results of our proposed PCDVQ are marked in gray.

| Methods | LLaMA-2-7B | | | | LLaMA-2-13B | | | | LLaMA-2-70B | | | |
|---|---|---|---|---|---|---|---|---|---|---|---|---|
| | bit | Wiki2↓ | C4↓ | QA Avg↑ | bit | Wiki2↓ | C4↓ | QA Avg↑ | bit | Wiki2↓ | C4↓ | QA Avg↑ |
| - | 16 | 5.12 | 6.63 | 62.24 | 16 | 4.57 | 6.05 | 65.38 | 16 | 3.12 | 4.97 | 70.21 |
| GPTQ | 2.125 | 50.75 | 36.76 | 39.16 | 2.125 | 43.84 | 23.07 | 43.72 | 2.125 | NaN | NaN | 59.18 |
| GPTVQ | 2.25 | 6.71 | 9.90 | 56.14 | 2.25 | 5.72 | 8.42 | 61.56 | 2.25 | 4.25 | 6.90 | 68.55 |
| DB-LLM | 2.01 | 7.23 | 9.62 | 55.12 | 2.01 | 6.19 | 8.38 | 59.41 | 2.01 | 4.64 | 6.77 | 65.83 |
| QuIP# | 2.02 | 6.19 | 8.16 | 58.23 | 2.00 | 5.35 | 7.20 | 61.96 | 2.00 | 3.91 | 5.71 | 68.94 |
| AQLM | 2.29 | 6.29 | 8.56 | 58.57 | 2.18 | 5.41 | 7.20 | 61.58 | 2.07 | 3.94 | 5.72 | 68.75 |
| VPTQ | 2.02 | 6.13 | 8.07 | 58.13 | 2.02 | 5.32 | 7.15 | 62.37 | 2.07 | 3.93 | 5.72 | 68.61 |
| | 2.26 | 5.95 | 7.87 | 59.36 | 2.18 | 5.28 | 7.04 | 63.11 | 2.11 | 3.92 | 5.71 | 68.69 |
| PCDVQ | 2.00 | 5.81 | 8.37 | 58.60 | 2.00 | 5.31 | 7.23 | 63.10 | 2.00 | 3.55 | 5.38 | 69.28 |
| | 2.25 | 5.68 | 7.79 | 60.44 | 2.25 | 5.04 | 6.90 | 63.66 | 2.25 | 3.41 | 5.23 | 69.74 |

Table 2: Quantization performance comparison on LLaMA-3 series and Mistral-7B. The context length of WikiText2 evaluation is 2048 for LLaMA-3 and 8192 for Mistral-7B. Full results are in A.3.

| Methods | LLaMA-3-8B | | | LLaMA-3-70B | | | Methods | Mistral-7B | | |
|---|---|---|---|---|---|---|---|---|---|---|
| | bit | Wiki2↓ | QA Avg↑ | bit | Wiki2↓ | QA Avg↑ | | bit | Wiki2↓ | QA Avg↑ |
| - | 16 | 6.14 | 68.66 | 16 | 2.90 | 75.32 | - | 16 | 6.02 | 68.61 |
| QuIP | 2 | 85.10 | 36.82 | 2 | 13 | 48.66 | GPTQ | 2.125 | 1535 | 44.45 |
| DB-LLM | 2 | 13.6 | 51.74 | NaN | NaN | NaN | QuIP# | 2 | 6.02 | 62.17 |
| GPTQ | 2 | 210 | 36.16 | 2 | 11.90 | 45.42 | AQLM | 2.01 | 6.32 | 62.19 |
| VPTQ | 2.08 | 9.29 | 60.22 | 2.02 | 5.60 | 70.86 | GPTVQ | 2.25 | 8.99 | 57.66 |
| | 2.24 | 9.19 | 62.68 | 2.07 | 5.66 | 70.74 | VPTQ | 2.04 | 5.64 | 63.20 |
| PCDVQ | 2 | 8.77 | 60.60 | 2 | 5.22 | 71.72 | PCDVQ | 2 | 5.53 | 63.85 |
| | 2.25 | 7.93 | 63.91 | 2.25 | 5.10 | 71.98 | | 2.25 | 5.38 | 64.33 |

## 4.3 ABLATION STUDY

**Ablation on Fine-tuning** Supervised fine-tuning has become a commonly utilized approach within VQ methods for enhancing the quantization performance. This work focuses on the sensitivity gap and the codebook construction, thereby we directly employ the related methods in QuIP#. Specifically, the block-wise fine-tuning adjusts the un-quantized weights of the current decoder block, and the e2e fine-tuning modifies the parameters of all normalization layers. We design to remove the tuning functions from the quantization methods as shown in Table 3. It can be observed that our PCDVQ can also outperforms QuIP# under the same settings, further demonstrating its effectiveness.

Table 3: Ablation study on the tuning metric. We perform 2-bit quantization experiments on LLaMA-2-7B and compare with QuIP#. The symbols 'w' and 'wo' denotes 'with' and 'without', respectively.

| Methods | w all tuning | | | wo block tuning | | | wo e2e tuning | | | wo all tuning | | |
|---|---|---|---|---|---|---|---|---|---|---|---|---|
| | Wiki2↓ | C4↓ | QA Avg↑ | Wiki2↓ | C4↓ | QA Avg↑ | Wiki2↓ | C4↓ | QA Avg↑ | Wiki2↓ | C4↓ | QA Avg↑ |
| - | 5.12 | 6.63 | 62.24 | 5.12 | 6.63 | 62.24 | 5.12 | 6.63 | 62.24 | 5.12 | 6.63 | 62.24 |
| QuIP# | 6.19 | 8.16 | 58.23 | 6.82 | 9.54 | 55.91 | 6.78 | 8.51 | 56.47 | 9.05 | 11.98 | 52.32 |
| PCDVQ | 5.81 | 8.37 | 58.6 | 6.60 | 9.22 | 58.73 | 6.61 | 8.36 | 59.52 | 8.47 | 10.92 | 55.89 |

**Ablation on PCD** To evaluate the effectiveness of the proposed PCD technique, a metric with the same unit for the direction and magnitude error is required. This is because the direction similarity is typically accessed by the cosine function, while that for magnitude can be various, such as Euclidean distance. To this end, we utilize the method shown in Figure 1 (b). Since the spatially vector error is $-v + c$ and the magnitude MSE is obviously $(\|v\| - \|c\|)^2$, the direction MSE cane be regarded to be $2\|v\|^2 (1 - \cos\theta)$ reasonably. Based on this, we collect the direction and magnitude MSE of QuIP# and our PCDVQ. Figure 3 demonstrates that our PCD significantly reduces the direction errors by averagely 0.3, verifying the necessity of the independent VQ and more bits for direction.

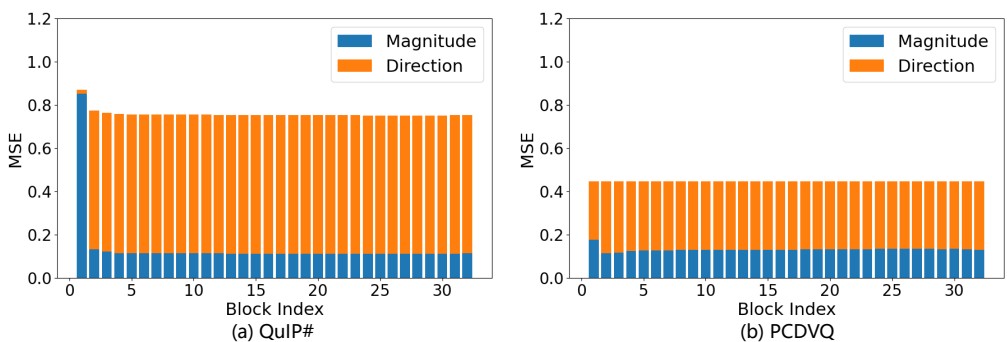

Figure 3: Ablation study on PCD. Both QuIP# and PCDVQ are in 2-bit setting. The x-axis denotes the indices of decoder blocks of LLaMA-2-7B. The y-axis shows the average MSE for vectors within each block. The measurement of MSE is same with Figure 1 (b).

**Ablation on DACC** To evaluate our proposed DACC, we introduce several typical methods for comparison: For the direction codebook, we construct the following codebooks. (1) Random Gaussian, which randomly samples directions from the standard Gaussian distribution. (2) Simulated Annealing, which aims to maximize the minimal cosine similarities across directions. (3) K-Means, which generates centers by directly clustering. For the magnitude codebook, we introduce the K-Means algorithm. It can be concluded from Table 4 that our approaches to construct codebooks for direction and magnitude parameters are the most effective.

Table 4: Ablation study on DACC. We perform 2.25-bit quantization experiments on LLaMA-2-7B. Our methods for direction and magnitude are denoted by Greedy $E_8$ and Llyod-Max.

| Direction | | | | | | | | Magnitude | | | |
|---|---|---|---|---|---|---|---|---|---|---|---|
| Random Gaussian | | Simulated Annealing | | K-Means | | Greedy $E_8$ | | K-Means | | Llyod-Max | |
| Wiki2↓ | QA Avg↑ | Wiki2↓ | QA Avg↑ | Wiki2↓ | QA Avg↑ | Wiki2↓ | QA Avg↑ | Wiki2↓ | QA Avg↑ | Wiki2↓ | QA Avg↑ |
| 2637.25 | 34.75 | 7.08 | 58.51 | 6.59 | 59.10 | 5.68 | 60.44 | 6.44 | 60.11 | 5.68 | 60.44 |

## 4.4 EFFICIENCY ANALYSIS

The PCDVQ (2-bit) can reduce approximately 87.5% of the memory consumption, and the performance of PCDVQ (2.25-bit) is 86.7%. Inherited from weight quantization, PCDVQ can also accelerate the inference process by minimizing the memory bandwidth. We evaluate the generation throughput on a NVIDIA RTX 4090 using the HuggingFace library's Llama implementation. The "tokens per second" increases from 33.1 to 95.7 for PCDVQ (2-bit).

## 5 CONCLUSION

In this paper, we focus on enhancing the accuracy of VQ methods. Our analysis reveals two key limitations for the current VQ paradigm. (1) Vector-grained codebooks quantize the direction and magnitude in a coupled manner, while direction is identified to be more sensitive to quantization than its magnitude counterpart. This huge gap is primarily caused by the dimensional difference. (2) The metric of accessing the vector similarity (Euclidean distance) that widely applied in existing VQ works is more sensitive to magnitude errors. This property is contrary to the above finding and can lead to larger quantization errors. To beyond these limitations, we propose PCDVQ, a novel VQ framework with enhanced accuracy and high compression ratio. Specifically, we introduce the polar coordinate decoupling to separately performing VQ and allocate more bits of index to the direction parameters. We also incorporate a distribution aligned codebook construction technique, which establishes a spatially uniform direction codebook and a root Chi-square magnitude distribution. Our proposed PCDVQ achieves the superior performance across six LLMs and seven tasks at the 2-bit level setting, demonstrating a more effective VQ methodology.

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

## A  APPENDIX

### A.1  THE USE OF LARGE LANGUAGE MODELS (LLMS)

During the completion of this manuscript, LLMs were employed solely to polish the writing, such as checking grammar and spelling errors, adjusting sentence structures, and refining the narrative style. We used LLMs only to a limited extent to enhance the quality of the article, and their role was insufficient to warrant listing them as major contributors.

### A.2  DERIVATION OF ROOT CHI-SQUARE DISTRIBUTION

Suppose there is a magnitude variable $X$ such that $Y = X^2$ follows a Chi-square distribution with degrees of freedom k. We need to derive the probability density function (PDF) and cumulative distribution function (CDF) of $X$. Firstly, the PDF of Chi-square distribution is:

$$f_Y(y) = \frac{1}{2^{\frac{k}{2}}\Gamma(\frac{k}{2})} y^{\frac{k}{2}-1} e^{-\frac{y}{2}}, y \geq 0. \tag{12}$$

Since $Y = X^2$ and $X$ is non-negative, the PDF of $X$ can be obtained using the variable transformation method. The transformation function is $Y = g(X) = X^2$, and its inverse function is $X = h(Y) = \sqrt{Y}$. According to the variable transformation formula:

$$f_Y(y) = f_X(h(y))|h'(y)|. \tag{13}$$

Given $h(y) = \sqrt{Y}$ and $h'(y) = \frac{1}{2\sqrt{y}}$, Eq. 13 can be transformed to:

$$f_Y(y) = f_X(\sqrt{y}) \cdot \frac{1}{2\sqrt{y}}. \tag{14}$$

Substituting $y$ with $x^2$, Eq. 14 can be expressed as:

$$f_Y(x^2) = f_X(x) \cdot \frac{1}{2x}. \tag{15}$$

Subsequently, the expression of $f_X(x)$ can be obtained:

$$f_X(x) = 2x \cdot f_Y(x^2). \tag{16}$$

Substituting Eq. 12 into Eq. 16, the PDF of $X$ should be:

$$f_X(x) = 2x \cdot \left(\frac{1}{2^{\frac{k}{2}}\Gamma(\frac{k}{2})}(x^2)^{\frac{k}{2}-1} e^{-\frac{x^2}{2}}\right) = \frac{2^{1-\frac{k}{2}} x^{k-1} e^{-\frac{x^2}{2}}}{\Gamma(\frac{k}{2})}, x \geq 0. \tag{17}$$

Considering that $Y = X^2$, the CDF of $X$ can be expressed as the value of the Chi-square distribution at $x^2$:

$$F_X(x) = P(X \le x) = P(Y \le x^2) = F_Y(x^2). \tag{18}$$

The symbol $F_Y$ denotes the CDF of the Chi-square distribution, which can be expressed by:

$$F_Y(y) = \frac{\gamma(\frac{k}{2}, \frac{y}{2})}{\Gamma(\frac{k}{2})}. \tag{19}$$

Thereby, the CDF of $X$ should be:

$$F_X(x) = \frac{\gamma(\frac{k}{2}, \frac{x^2}{2})}{\Gamma(\frac{k}{2})}, x \ge 0. \tag{20}$$

### A.3 DETAILED RESULTS OF ZERO-SHOT EVALUATION

We perform the common QA evaluation on five zero-shot datasets, including Arc-Challenge Clark et al. (2018), Arc-Easy Clark et al. (2018), HellaSwag Zellers et al. (2019), PIQA Bisk et al. (2020), and WinoGrande Sakaguchi et al. (2021). Tables in the Experiments section exhibit the average performance for representation. We here display the full results on each dataset.

Table 5: Full results of zero-shot evaluation on LLaMA-2-7B.

| Methods | | LLaMA-2-7B | | | | | |
| | bit | Arc-Challenge↑ | Arc-Easy↑ | HellaSwag↑ | PIQA↑ | WinoGrande↑ | QA Avg↑ |
| --- | --- | --- | --- | --- | --- | --- | --- |
| - | 16 | 39.93 | 69.28 | 56.69 | 78.35 | 66.93 | 62.24 |
| GPTQ | 2.125 | 20.90 | 34.90 | 30.50 | 57.20 | 52.30 | 39.16 |
| GPTVQ | 2.25 | 31.20 | 66.30 | 46.40 | 72.40 | 64.40 | 56.14 |
| DB-LLM | 2.01 | 33.53 | 45.20 | **61.98** | 73.18 | 61.72 | 55.12 |
| QuIP# | 2 | 34.60 | 64.60 | 51.91 | 75.14 | 64.90 | 58.23 |
| AQLM | 2.29 | 34.90 | 66.5 | 50.88 | 74.92 | 65.67 | 58.57 |
| VPTQ | 2.02 | 35.24 | 63.80 | 52.08 | 75.19 | 64.33 | 58.13 |
| | 2.26 | 36.43 | 64.90 | 52.87 | 76.17 | **66.46** | 59.36 |
| PCDVQ | 2 | 37.20 | 64.40 | 50.71 | 75.40 | 65.74 | 58.60 |
| | 2.25 | **38.31** | **67.92** | 53.13 | **76.38** | **66.46** | **60.44** |

Table 6: Full results of zero-shot evaluation on LLaMA-2-13B.

| Methods | | LLaMA-2-13B | | | | | |
| | bit | Arc-Challenge↑ | Arc-Easy↑ | HellaSwag↑ | PIQA↑ | WinoGrande↑ | QA Avg↑ |
| --- | --- | --- | --- | --- | --- | --- | --- |
| | 16 | 45.56 | 73.23 | 59.71 | 78.73 | 69.69 | 65.38 |
| GPTQ | 2.125 | 23.30 | 43.30 | 36.00 | 61.30 | 54.70 | 43.72 |
| GPTVQ | 2.25 | 38.70 | 73.60 | 51.60 | 75.40 | 68.50 | 61.56 |
| DB-LLM | 2.01 | 38.14 | 51.64 | **68.04** | 75.14 | 64.09 | 59.41 |
| QuIP# | 2 | 39.50 | 69.30 | 56.01 | 77.30 | 67.70 | 61.96 |
| AQLM | 2.18 | 39.42 | 69.15 | 54.68 | 76.22 | 68.43 | 61.58 |
| VPTQ | 2.02 | 40.02 | 71.55 | 56.18 | 77.26 | 66.85 | 62.37 |
| | 2.18 | 40.96 | **71.80** | 56.89 | **77.48** | 68.43 | 63.11 |
| PCDVQ | 2 | 43.00 | 71.21 | 54.66 | 76.98 | 69.69 | 63.10 |
| | 2.25 | **43.34** | 70.53 | 56.57 | 77.25 | **70.63** | **63.66** |

Table 5 6 7 are full results of the Table 1. Table 8 9 10 are full results of the Table 2. QA Avg is the average result of the above five zero-shot evaluation tasks. The best performance for each task is highlighted in bold, and results of our proposed PCDVQ are marked in gray.

It can be observed that our proposed PCDVQ achieves the best average accuracy, although there are minimal gaps between baselines on few datasets and models.

Table 7: Full results of zero-shot evaluation on LLaMA-2-70B.

| Methods | bit | Arc-Challenge↑ | Arc-Easy↑ | HellaSwag↑ | PIQA↑ | WinoGrande↑ | QA Avg↑ |
|---|---|---|---|---|---|---|---|
| | 16 | 51.11 | 77.74 | 63.97 | 81.12 | 77.11 | 70.21 |
| GPTQ | 2.125 | 35.80 | 67.00 | 51.80 | 74.60 | 66.70 | 59.18 |
| GPTVQ | 2.25 | **49.40** | **80.47** | 58.26 | 79.40 | 75.20 | 68.55 |
| DB-LLM | 2.01 | 44.45 | 55.93 | 76.16 | 79.27 | 73.32 | 65.83 |
| QuIP# | 2 | 48.70 | 77.30 | 62.49 | 80.30 | 75.90 | 68.94 |
| AQLM | 2.07 | 47.93 | 77.68 | 61.79 | 80.43 | 75.93 | 68.75 |
| VPTQ | 2.07 | 47.7 | 77.10 | 62.98 | 80.30 | 74.98 | 68.610 |
| | 2.11 | 48.29 | 77.70 | 62.51 | 79.82 | 75.14 | 68.69 |
| PCDVQ | 2 | 48.03 | 77.56 | 62.43 | 81.33 | 77.03 | 69.28 |
| | 2.25 | 48.97 | 77.69 | **62.99** | **81.33** | **77.74** | **69.74** |

Table 8: Full results of zero-shot evaluation on LLaMA-3-8B.

| Methods | bit | Arc-Challenge↑ | Arc-Easy↑ | HellaSwag↑ | PIQA↑ | WinoGrande↑ | QA Avg↑ |
|---|---|---|---|---|---|---|---|
| | 16 | 50.30 | 80.10 | 60.20 | 79.60 | 73.10 | 68.66 |
| QuIP | 2 | 21.30 | 29.00 | 29.20 | 52.90 | 51.70 | 36.82 |
| DB-LLM | 2 | 28.20 | 59.10 | 42.10 | 68.90 | 60.40 | 51.74 |
| GPTQ | 2 | 19.90 | 28.80 | 27.70 | 53.90 | 50.50 | 36.16 |
| VPTQ | 2.08 | 36.90 | 71.00 | 52.20 | 75.10 | 65.90 | 60.22 |
| | 2.24 | 42.60 | 73.20 | 53.10 | 75.40 | 69.10 | 62.68 |
| PCDVQ | 2 | 37.54 | 71.75 | 51.57 | 74.59 | 67.56 | 60.60 |
| | 2.25 | **43.68** | **75.04** | **54.29** | **76.49** | **70.08** | **63.91** |

Table 9: Full results of zero-shot evaluation on LLaMA-3-70B.

| Methods | bit | Arc-Challenge↑ | Arc-Easy↑ | HellaSwag↑ | PIQA↑ | WinoGrande↑ | QA Avg↑ |
|---|---|---|---|---|---|---|---|
| | 16 | 60.10 | 87.00 | 66.30 | 82.40 | 80.80 | 75.32 |
| QuIP | 2 | 26.50 | 48.90 | 40.90 | 65.30 | 61.70 | 48.66 |
| DB-LLM | NaN | NaN | NaN | NaN | NaN | NaN | NaN |
| GPTQ | 2 | 24.60 | 38.90 | 41.00 | 62.70 | 59.90 | 45.42 |
| VPTQ | 2.02 | 52.50 | 81.80 | 61.70 | 80.40 | 77.90 | 70.86 |
| | 2.07 | 54.20 | 83.60 | 61.80 | 80.10 | 74.00 | 70.74 |
| PCDVQ | 2 | 54.63 | 83.5 | 61.92 | **80.8** | 77.78 | 71.72 |
| | 2.25 | **55.02** | **83.71** | **62.24** | 80.75 | **78.22** | **71.98** |

Table 10: Full results of zero-shot evaluation on Mistral-7B.

| Methods | bit | Arc-Challenge↑ | Arc-Easy↑ | HellaSwag↑ | PIQA↑ | WinoGrande↑ | QA Avg↑ |
|---|---|---|---|---|---|---|---|
| | 16 | 48.89 | 78.87 | 61.12 | 80.3 | 73.88 | 68.61 |
| GPTQ | 2.125 | 24.49 | 44.91 | 36.56 | 63.33 | 52.96 | 44.45 |
| QuIP# | 2 | 39.76 | 72.14 | 52.95 | 76.71 | 69.30 | 62.17 |
| AQLM | 2.01 | 40.44 | **73.65** | 52.13 | 76.01 | 68.75 | 62.19 |
| GPTVQ | 2.25 | 37.37 | 71.00 | 45.43 | 70.18 | 64.33 | 57.66 |
| VPTQ | 2.04 | 41.13 | 72.22 | 56.10 | 77.91 | 68.67 | 63.20 |
| PCDVQ | 2 | 41.66 | 73.51 | 56.9 | 78.07 | 69.12 | 63.85 |
| | 2.25 | **42.20** | 72.88 | **57.39** | **78.52** | **70.70** | **64.33** |

### A.4 BIT SETTING

The bit here denotes the Bits Per Weight (BPW). After quantization, the vector is represented by a index of direction and a index of magnitude. For PCDVQ at 2.25-bit, we set a to 16 and b to 2. As we set the vector dimension $k$ to 8, the BPW can be calculated by $(a + b)/k = 2.25$. For PCDVQ at 2.25-bit, the BPW can be calculated by $(14 + 2)/8 = 2.00$. Notably, the bits consumption of codebooks are shared by the total model with billions of positions, which is minimal enough to be omitted.

