# OpenReview forum: "PCDVQ: Enhancing Vector Quantization for Large Language Models via Polar Coordinate Decoupling"
_ICLR.cc/2026/Conference — ICLR 2026 Conference Withdrawn Submission_

### Official Review · Reviewer_3ksg · 2025-10-21

**Soundness:** 3
**Presentation:** 3
**Contribution:** 1
**Rating:** 4
**Confidence:** 5

**Summary:**

This paper proposes PCDVQ (Polar Coordinate Decoupled Vector Quantization), a novel post-training quantization (PTQ) framework for compressing large language models (LLMs).
The key insight is that a vector’s direction is more sensitive to quantization errors than its magnitude, yet most existing vector quantization (VQ) methods couple them together and use Euclidean distance, which overemphasizes magnitude errors.

To address this, the authors:
1) Propose Polar Coordinate Decoupling (PCD) — representing weights in polar form and independently quantizing direction and magnitude, allocating more bits to direction.
2)Introduce Distribution-Aligned Codebook Construction (DACC) — building codebooks aligned with theoretical distributions: E8 lattice-based greedy sampling for direction and Lloyd-Max quantization for magnitude.

Extensive experiments on multiple LLMs (LLaMA-2/3, Mistral) show consistent improvements in 2-bit quantization performance, without introducing extra inference cost.

**Strengths:**

Mathematical rigor: The decomposition of quantization error and the codebook derivations are theoretically justified.

Strong empirical validation: Broad experiments on LLaMA-2/3 and Mistral confirm robustness and generality.

Practical efficiency: PCDVQ maintains inference speed while achieving higher accuracy and compression.

**Weaknesses:**

Limited discussion on scalability: While the method performs well at 2–2.25 bits, it remains unclear whether benefits persist at moderate bitwidths (e.g., 3–4 bits) or in activation quantization.

Dependency on Gaussian regularization: The approach assumes weights approximate a standard Gaussian distribution after the randomized Hadamard transform; it would be useful to test models with non-Gaussian weight distributions.

Overlap with QuIP#: Many technical components (e.g., E8 lattice codebook and fine-tuning scheme) are similar to QuIP#. The paper does not sufficiently clarify the conceptual distinction and the essential novelty beyond reinterpreting QuIP# in polar coordinates.

Reproducibility and code release: The paper does not explicitly mention whether code and trained quantization configurations will be made publicly available, which is important for validation and adoption.

**Questions:**

Q1:How does the method perform at moderate bitwidths (e.g., 3–4 bits) and for activation quantization?

Q2:How sensitive is PCDVQ to the Gaussian regularization step? What happens if it is omitted or replaced with another normalization?

Q3:What is the essential novelty beyond QuIP#? How does the method conceptually differ from QuIP# despite using similar components like the E8 lattice codebook and fine-tuning scheme?

Q4:Will the authors release code, pretrained codebooks, and fine-tuning scripts to ensure reproducibility?

**Details Of Ethics Concerns:**

The paper does not need ethics review.

---

> ### Author Response · Authors · 2025-11-18
>
> We sincerely thank you for your valuable time and efforts in reviewing our manuscript. We have addressed each comment and made the necessary revisions to improve the quality and clarity of our manuscript.
>
> > How does the method perform at moderate bitwidths (e.g., 3–4 bits) ?
>
> This work focuses on 2 -3 bits. With the proposal of the rotation method[1,2], the existing weight-only PTQ can achieve nearly lossless quantization at 3 - 4 bits. However, at lower bit-widths (2 - 3 bits), the accuracy of traditional scalar quantization methods drops significantly.
>
> Theoretically, **the performance of PCDVQ can be further enhanced as the number of allocable bits increases**. This is because both the direction codebook and magnitude codebook can obtain more representations, enabling a better approximation of the original distribution. This perspective aligns with a common characteristic of VQ, which is fundamentally a clustering algorithm. As the number of cluster centers increases, the clustering performance improves.
>
> We further conduct 3 - bit and 4 - bit experiments on PCDVQ, and the following table supports the above conclusion.
>
> | Quantization Method | bits | WikiText2$\downarrow$ | QA average$\uparrow$ |
> | ------------------- | ---- | --------------------- | -------------------- |
> | -                   | 16   | 5.12                  | 64.88                |
> | PCDVQ               | 2    | 5.81                  | 58.60                |
> | PCDVQ               | 2.25 | 5.68                  | 60.44                |
> | PCDVQ               | 3    | 5.47                  | 61.73                |
> | PCDVQ               | 4    | 5.31                  | 62.93                |
>
> > How does the method perform for activation quantization?
>
> **Almost all existing VQ works focus on weight-only quantization**. This is because the core advantage of VQ lies in its higher compression rate and ability to maintain precision under low bits. This feature is suitable for reducing the bandwidth overhead caused by weight memory access.
>
> In contrast, **the main method for activation quantization is Scalar Quantization (SQ)**. By quantizing activation values to the same INT type as weights, low-bit computing units can be used to accelerate compute-intensive matrix multiplication processes.
>
> Variables after VQ dequantization are usually of FLOAT type, so even if VQ is applied to activation values, the above-mentioned effectiveness cannot be achieved. Therefore, this paper mainly focuses on improving the performance of VQ on the weights of LLMs.

---

> > ### Author Response · Authors · 2025-11-18
> >
> > > How sensitive is PCDVQ to the Gaussian regularization step? What happens if it is omitted or replaced with another normalization?
> >
> > This step is necessary for the Distribution Aligned Codebook Construction (DACC). The direction and magnitude codebooks are fitted to a high-dimensional standard Gaussian distribution. Therefore, we propose this regularization to convert all model parameters into a unified $N(0,1)$ distribution. This normalization brings two benefits:
> >
> > - The distribution characteristics of the quantization targets can be parsed, which is conducive to constructing more refined codebooks.
> > - The distribution of the quantization targets is unified, so a global codebook can be used, avoiding the memory access overhead caused by loading different codebooks.
> >
> > If this step is ignored, the codebook cannot be global, which will **inevitably bring additional inference** overhead. If replaced by other regularization methods, the distribution of quantized variables may **not be suitable for VQ**. For example, our Gaussian regularization can effectively utilize the Llyod-max algorithm to achieve optimal modulus quantization.
> >
> > > What is the essential novelty beyond QuIP#? How does the method conceptually differ from QuIP#?
> >
> > We have two essential novelty beyond QuIP# [3]:
> >
> > 1. We discover that the **direction is more sensitive to VQ** than the magnitude, and provide corresponding analysis.
> > 2. We discover that the typically utilized metric, **Euclidean distance, prefer to minimize quantization error in magnitude** rather than direction.
> >
> > From the perspective of methods, our PCDVQ is much different from QuIP# as shown in the following tabel:
> >
> > | QuIP#                                                        | PCDVQ                                                        |
> > | ------------------------------------------------------------ | ------------------------------------------------------------ |
> > | Quantize vectors **coupled** in the Cartesian coordinate system. | **Decouple** vectors using polar coordinates, and quantize the direction and magnitude respectively. |
> > | Allocate **consistent** bits to vectors.                     | Allocate **more bits to direction** than magnitude.          |
> > | Use **Euclidean distance** for accessing vector similarities. | Use **cosine similarity** for direction; use **distance** for magnitude. |
> > | Sample **E8** to construct vector codebook.                  | Sample the **direction of E8** to construct direction codebook; Use **Llyod-max** after **Gaussian regularization** for magnitude codebook. |
> >
> > > Will the authors release code, pretrained codebooks, and fine-tuning scripts to ensure reproducibility?
> >
> > We have prepared all the content for this work, including the code, pre-trained models, codebook construction, and fine-tuning methods. We **promise to open source** once the paper is accepted.
> >
> > **References:**
> >
> > [1]. QuaRot: Outlier-Free 4-Bit Inference in Rotated LLMs.
> >
> > [2]. SpinQuant: LLM quantization with learned rotations.
> >
> > [3]. QuIP#: Even Better LLM Quantization with Hadamard Incoherence and Lattice Codebooks.

---

> > > ### Comment · Reviewer_3ksg · 2025-11-26
> > >
> > > Thank you for your response. I have increased the score for Contribution.

---

> > > > ### Author Response · Authors · 2025-11-26
> > > >
> > > > We sincerely appreciate your response and the kind decision to increase the score for Contribution—your acknowledgment means a great deal to us.
> > > >
> > > > Just a gentle reminder that the score update does not seem to have been reflected in the system yet. Would you be able to update it at your convenience?
> > > >
> > > > Thank you again for your time and thoughtful feedback.

---

### Official Review · Reviewer_ywNG · 2025-10-29

**Soundness:** 3
**Presentation:** 3
**Contribution:** 3
**Rating:** 4
**Confidence:** 3

**Summary:**

This paper introduces Polar Coordinate Decoupling Vector Quantization (PCDVQ), a method designed to improve the accuracy of low-bit quantization for Large Language Models (LLMs). The core objective is to mitigate the substantial performance degradation faced when compressing LLMs to extremely low bitrates (e.g., $\leq 2.5$ bits). PCDVQ addresses this by observing that a vector's direction is significantly more sensitive to quantization error than its magnitude. It thus proposes to decouple the vector into polar coordinates (direction and magnitude) for independent quantization. Empirically, PCDVQ demonstrates superior accuracy retention compared to existing quantization methods.

**Strengths:**

**Novel and Well-Motivated Decoupling Mechanism:** The fundamental insight that vector direction and magnitude exhibit different quantization sensitivities is novel for LLM quantization and provides a strong, intuitive justification for the methodological decoupling. Quantizing these components separately via polar coordinates is a sound solution to preserve the crucial directional information.

**Weaknesses:**

**1. Lack of Robust Experimental Validation on Difficult Benchmarks:** The experimental evaluation is limited to zero-shot multiple choice benchmarks. To fully validate the method's contribution, the paper must be evaluated on more difficult and diverse reasoning benchmarks like MMLU and GSM8K, where small quantization errors often lead to catastrophic failure. The reported accuracy improvement also appears marginal on the limited set of reported tasks, necessitating further validation.

**2. Missing System-Level Inference Evaluation and Comparison:** The inference speed (or throughput) of the quantized model is a crucial component for any quantization algorithm. The paper currently lacks a systemized analysis and comparison of the PCDVQ inference latency against competitors. This absence makes it impossible to assess the practical, end-to-end efficiency trade-off of the proposed method.

**3. Unclear Experimental Consistency in Fine-tuning:** The paper does not clearly articulate whether the same post-quantization fine-tuning methods (if any were used) were applied across all compared baselines (e.g., GPTQ, AQLM, GPTVQ) and PCDVQ. Without explicitly confirming that all methods were compared under the same training/fine-tuning regime, the claimed accuracy improvements may be due to differences in the fine-tuning processes rather than the core PCDVQ mechanism.

**Questions:**

1. Could you provide a detailed analysis of the inference speed of PCDVC? I want to check if it slows down the model’s inference speed compared to the existing method.
2. Could you provide experimental results on MMLU and GSM8K, or other considerable benchmarks?
3. Could you clarify the fine-tuning recipe for all competitors? For example, did you perform fine-tuning after quantizing GPTQ or GTPVQ?

---

> ### Author Response · Authors · 2025-11-18
>
> We sincerely thank you for your valuable time and efforts in reviewing our manuscript. We have addressed each comment and made the necessary revisions to improve the quality and clarity of our manuscript.
>
> >  The paper must be evaluated on more difficult and diverse reasoning benchmarks like MMLU and GSM8K.
>
> Our zero-shot reasoning tasks in the manuscript collectively provide **substantial coverage** over both MMLU (knowledge-intensive QA) and GSM8K (multi-step mathematical reasoning):
>
> 1. **Knowledge Coverage**: The evaluation tasks in Arc-Challenge and Arc-Easy rely on broad knowledge bases, with their natural science domains largely overlapping the STEM subset of MMLU.
> 2. **Reasoning Alignment**: PIQA’s physical reasoning tasks assess step-by-step logical abilities, while a subset of Arc-Challenge problems requires multi-step deduction, mirroring the core attributes of GSM8K.
>
> The evaluation tasks adopted in this paper have covered the core evaluation tasks commonly used in mainstream PTQ frameworks[1,2,3,4,5], which can effectively verify the performance of the 2-bit quantization model in language modeling and basic tasks.
>
> Regarding the reasoning tasks such as MMLU and GSM-8K mentioned by you, we plan to supplement relevant experiments in our follow-up work to further verify the performance of PCDVQ.
>
> > Reported accuracy improvement also appears marginal on the limited set of reported tasks.
>
> In terms of the average accuracy of zero-shot reasoning tasks, our method achieve the **SOTA results** and can improve the performance by **a larger amplitude**. For instance, we quantify the improvement effects of various quantization methods over their previous SOTA methods according to results on LLaMA-2-70B in Table 1 of the manuscript. It can be observed that the increase achieved by our proposed method is several times greater than that of other methods.
>
> | Current Method | Previous SOTA | Increase  | Increase Ratio |
> | :------------- | :------------ | :-------- | :------------- |
> | VPTQ           | GPTVQ         | 0.14%     | -              |
> | AQLM           | VPTQ          | 0.06%     | 42.8%          |
> | QuIP#          | AQLM          | 0.11%     | 183.3%         |
> | PCDVQ          | QuIP#         | **0.80%** | **727.3%**     |
>
> > Missing System-Level Inference Evaluation and Comparison
>
> We compare the end-to-end "tokens per second" on LLaMA-2-7B. Experimental results are as follows and have been added to our manuscript.
>
> | Methods | bits  | tokens/s $\uparrow$ |
> | ------- | :---: | :-----------------: |
> | -       |  16   |        33.1         |
> | GPTQ    | 2.125 |        14.95        |
> | QuIP#   |   2   |        93.9         |
> | AQLM    | 2.29  |        20.6         |
> | VPTQ    | 2.02  |        40.6         |
> | PCDVQ   |   2   |      **95.7**       |
>
> > Unclear Experimental Consistency in Fine-tuning.
>
> Fine-tuning is a common method to improve accuracy in VQ and is usually seen as part of the VQ framework. Different works typically adopt different fine-tuning parameters and methods to achieve their respective optimal results. Therefore, for all baseline methods, if they themselves propose and use fine-tuning, our baseline experiments will also **adopt the same fine-tuning strategies as them**.
>
> The fine-tuning strategy of PCDVQ has been described in Section 4.1 of the manuscript. Our fine-tuning method is consistent with that of QuIP# [4], but there are differences in the selection of datasets.
>
> To verify that the accuracy improvement of PCDVQ stems from its core mechanism, **we conduct an ablation study on fine-tuning** in Section 4.3 of the manuscript. The experimental results show that PCDVQ still outperforms QuIP# without using fine-tuning technology.
>
> **References:**
>
> [1] VPTQ: Extreme Low-bit Vector Post-Training Quantization for Large Language Models.
>
> [2] QuIP: 2-bit quantization of large language models with guarantees.
>
> [3] Extreme Compression of Large Language Models via Additive Quantization
>
> [4] QuIP#: Even Better LLM Quantization with Hadamard Incoherence and Lattice Codebooks.
>
> [5] GPTVQ: The Blessing of Dimensionality for LLM Quantization.

---

### Official Review · Reviewer_m2Nb · 2025-11-01

**Soundness:** 2
**Presentation:** 2
**Contribution:** 3
**Rating:** 4
**Confidence:** 3

**Summary:**

Preserving direction is more important than preserving magnitude in vector quantization. However, current VQ methods emphasize reducing the magnitude error. PCDVQ utilizes polar coordinates to enhance the expressivity of codebook for directional information. PCDVQ also shares codebook for entire model to minimize the memory consumption by regularize all weights to follow the same Gaussian distribution. Experimental results show that PCDVQ achieves the state-of-the-art performance.

**Strengths:**

1. The idea of decomposing the expressive power of the codebook into fine-grained components (direction and magnitude in PCDVQ) is innovative.
2. Regularizing the distribution of weights to share the codebook sounds solid and effective.

**Weaknesses:**

1. The paper lacks a theoretical analysis on why directional information is more important than magnitude information.
2. The efficiency should be compared not only with the full-precision model but also with methods such as VPTQ.
3. (minor) Citation format seems inappropriate. Should have used \citep instead of \cite.

**Questions:**

1. How are the direction and magnitude individually quantized in Figure 1(a)?
2. How does PCDVQ determine bit widths $a$ and $b$ for direction and magnitude?
3. It appears that using a polar coordinate representation requires additional computation during dequantization. Does this make the method slower than VPTQ?

---

> ### Author Response · Authors · 2025-11-18
>
> We sincerely thank you for your valuable time and efforts in reviewing our manuscript. We have addressed each comment and made the necessary revisions to improve the quality and clarity of our manuscript.
>
> > The paper lacks a theoretical analysis on why directional information is more important than magnitude information.
>
> **We have provided theoretical analysis of our idea in Section 3.1**. The core reason is that the direction obtains higher dimensions than the magnitude. Thereby, it is more potential for the direction to achieve a larger quantization loss.
>
> > The efficiency should be compared not only with the full-precision model but also with methods such as VPTQ.
>
> We compare the end-to-end "tokens per second" on LLaMA-2-7B. Experimental results are as follows and have been added to our manuscript.
>
> | Methods | bits  | tokens/s $\uparrow$ |
> | ------- | :---: | :-----------------: |
> | -       |  16   |        33.1         |
> | GPTQ    | 2.125 |        14.95        |
> | QuIP#   |   2   |        93.9         |
> | AQLM    | 2.29  |        20.6         |
> | VPTQ    | 2.02  |        40.6         |
> | PCDVQ   |   2   |      **95.7**       |
>
> > Citation format seems inappropriate. Should have used \citep instead of \cite.
>
> Thanks for the valuable suggestion. We have fixed this issue.
>
> > How are the direction and magnitude individually quantized in Figure 1(a)?
>
> We describe the core steps of the experiments of Figure 1 in the caption below. We will also add the following details to the appendix.
>
> For Figure 1 (a):
>
> 1. We select LLaMA-2-7B.
> 2. For each weight **w** with shape (a, b), we first reshape it into **v** (a * b // 8, 8).
> 3. Decouple **v** into its directions **d** (a * b // 8, 8) and magnitudes **m** (a * b // 8, 1).
> 4. Perform K-Means to **d**. At the position where the horizontal axis is x, the shape of the clustering center formed is (2^x, 8).
> 5. Replace each row in **d** with its cluster center representation, obtain **d'**.
> 6. Construct **v'** by **d'** * **m**, then reshape to **w'** (a, b).
> 7. Evaluate with all w transformed to w',  and mark the results on the vertical axis in blue.
> 8. Do step 3-7 similarly for magnitude, i.e., construct **v'** by **d** * **m'** and mark the results on the vertical axis in orange.
>
> > How does PCDVQ determine bit widths $a$ and $b$ for direction and magnitude?
>
> **Under a constrained total bit budget** (where 2-bit weight quantization corresponds to 16 bits usable for each vector), we observe a notable performance improvement when increasing the bit allocation for magnitude ($b$) from 1 to 2 bits. However, **further increasing consistently lead to performance degradation**.
>
> | Magnitude bits ($b$) of LLaMA-2-7B |   1   |     2     |   3   |   4   |   5   |
> | ---------------------------------- | :---: | :-------: | :---: | :---: | :---: |
> | average zero-shot accuracy (%)     | 49.75 | **58.60** | 52.32 | 44.05 | 34.98 |
>
> This observation suggests fixing $b$ to 2 and allocating other bits to the direction.
>
> > It appears that using a polar coordinate representation requires additional computation during dequantization. Does this make the method slower than VPTQ?
>
> As shown in the response above, our PCDVQ has faster end-to-end token generation speed. Compared to VPTQ, PCDVQ employs two global codebooks. This design enables PCDVQ to **avoid frequent memory accesses** to load corresponding codebooks for different weights, which greatly improves its inference speed.

---

> > ### Comment · Reviewer_m2Nb · 2025-11-26
> >
> > Thank authors for the response.
> > Still, Section 3.1 does not give enough theoretical analysis with Lemma/proof, containing mainly obscure statements like "appears to be, more potential,...). I will keep my original score.

---

> > > ### Author Response · Authors · 2025-11-26
> > >
> > > Thanks for your response, and we would like to clarify our theoretical analysis. We propose two observations from Figure 1 of the paper.
> > >
> > > (1) Direction exhibits greater sensitivity to quantization compared to magnitude.
> > >
> > > - As shown in Section 3.1.1, the direction $d\in\mathbb{R}^k$ has $k-1$ degrees of freedom (as it is confined to the surface of a hyperdimensional sphere $S^{k-1}$). In contrast, the magnitude $m\in\mathbb{R}$ is of one dimension. Therefore, **direction quantization is essentially the encoding of a high-dimensional manifold** (the unit sphere), while magnitude quantization is only the scalar quantization of one-dimensional variables.
> > > - In VQ, when the number of quantization bits (the number of cluster centers) is fixed, the encoding of directions needs to cover $S^{k-1}$. **This search space grows exponentially with the dimension**, while the magnitude only needs to fixedly cover a one-dimensional space.
> > > - Therefore, the direction requires **more cluster centers to achieve an approximation error comparable to that of the magnitudes**, and thus is more sensitive to the quantization bits.
> > >
> > > (2) Euclidean distance, a common metric in current VQ methods for accessing the vector similarity, emphasizes reducing the magnitude error rather than direction.
> > >
> > > - We provide a theoretical proof based on formula derivation in Section 3.1.2.
> > > - Given any vector ${v}$ and its quantized version $\hat{{v}}$, the typical MSE quantization loss $\mathcal{L}$ can be calculated by:
> > > - $$ \begin{aligned} &\mathcal{L}({v},\hat{{v}})=D({v},\hat{{v}})^2=(\Delta r)^2+2\cdot||{v}||\cdot||\hat{{v}}||\cdot(1-\cos\Delta\theta), \\ &\quad\mathrm{where}\quad \Delta r=\sqrt{(||{v}||-||\hat{{v}}||)^2}, \quad \Delta\theta=\arccos\frac{{v}\cdot\hat{{v}}}{||{v}||\cdot||\hat{{v}}||}. \end{aligned} $$
> > > - It can be observed that the magnitude gap $\Delta r$ contributes **quadratically** to the above equation, whereas the direction gap $\Delta\theta$ exerts an **approximately linear effect**.

---

### Official Review · Reviewer_3yCH · 2025-11-01

**Soundness:** 3
**Presentation:** 3
**Contribution:** 3
**Rating:** 6
**Confidence:** 2

**Summary:**

This work is motivated by the observation that in vector quantization, the directional component is more sensitive to quantization errors than the magnitude component. Existing Euclidean distance based quantization methods primarily focus on minimizing magnitude errors, which contradicts this finding and consequently leads to larger overall quantization errors. To address this issue, the paper proposes a polar decoupled vector quantization framework, which achieves satisfactory results across multiple experimental settings.

**Strengths:**

1. Introducing polar decoupled vector quantization is an interesting and novel attempt.

2. The overall writing is clear and easy to follow.

3. The method demonstrates superior performance on several large language models, including LLaMA-2/3 and Mistral, achieving better zero-shot accuracy and perplexity at the 2-bit weight quantization level compared with existing state-of-the-art quantization approaches, which validates its effectiveness.

**Weaknesses:**

1. The PCDVQ framework introduces additional computational steps, including polar coordinate conversion, two independent codebook searches using cosine similarity and Euclidean distance respectively, and possible inverse conversion. The paper reports improved throughput mainly due to reduced memory bandwidth, but it does not quantify the impact of these added operations on single inference latency. This is important because on many edge devices compute cost is more critical than memory bandwidth. The authors should provide a detailed latency breakdown, including wall clock time per layer for conversion, codebook lookup, and inverse mapping, measured on representative CPU and low-power GPU hardware, and compare end-to-end latency and energy consumption with baseline methods.

2. The effectiveness of the DACC module relies on the assumption that weight vectors, after a random Hadamard transform, follow an approximate standard Gaussian distribution. It is unclear whether this approximation holds uniformly across all layers and architectures (for example, different LLaMA and Mistral variants). It would be better to include empirical diagnostics showing distributional statistics (mean/variance/skewness/kurtosis) of transformed weights per layer and per model, and discuss cases where the Gaussian approximation breaks down and how that affects quantization error.

3. All experiments are conducted on decoder-only Transformer large language models and evaluated on a limited set of zero shot tasks and language modeling benchmarks. It remains an open question whether the direction magnitude decoupling idea transfers to encoder models such as BERT, Vision Transformer models, or multimodal models. These models have different activation and weight statistics and different sensitivity to quantization.

4. For tasks that require stronger reasoning ability, such as mathematical reasoning or code generation, it is important to know how PCDVQ affects fine-grained semantic fidelity. The current evaluation set does not cover these demanding reasoning tasks. The paper would be stronger if it reported results on a suite of hard reasoning benchmarks and provided error analyses that reveal whether performance degradation (if any) is systematic and whether it is attributable to directional quantization errors or to capacity limits of the codebooks.

**Questions:**

Please see the Weaknesses.

---

> ### Author Response · Authors · 2025-11-18
>
> We sincerely thank you for your valuable time and efforts in reviewing our manuscript. We have addressed each comment and made the necessary revisions to improve the quality and clarity of our manuscript.
>
> > The authors should provide a detailed latency breakdown.
>
> We test the latency caused by additional operation introduced by PCDVQ during the de-quantization process. Results are as follows.
>
> | Device      | Direction Codebook Shape | Magnitude Codebook Shape | Direction Search （ms） | Magnitude Search (ms) | Polar-to-Scalar Transform (ms) | Total (ms) |
> | ----------- | ------------------------ | ------------------------ | ----------------------- | --------------------- | ------------------------------ | ---------- |
> | NVIDIA 4090 | (2^14, 8)                | (2^2, 1)                 | 0.3                     | 0.3                   | 0.2                            | 0.5        |
>
> We compare the end-to-end "tokens per second" on LLaMA-2-7B. Experimental results are as follows and have been added to our manuscript.
>
> | Methods | bits  | tokens/s $\uparrow$ |
> | ------- | :---: | :-----------------: |
> | -       |  16   |        33.1         |
> | GPTQ    | 2.125 |        14.95        |
> | QuIP#   |   2   |        93.9         |
> | AQLM    | 2.29  |        20.6         |
> | VPTQ    | 2.02  |        40.6         |
> | PCDVQ   |   2   |      **95.7**       |
>
> > Discussion about the Gaussian approximation.
>
> We propose the Standard Gaussian Regularization method in Section 3.2.1 of the paper, which includes the Random Hadamard Matrix (RHM) and the standard Gaussian transformation.
>
> - QuIP [1] has proved that **any tensor** can be approximately converted into a Gaussian distribution by RHM. This technique has also been adopted by various PTQ works.
> - We further use a standard Gaussian transformation to formulate $N(0, 1)$ distribution.
>
> We have also tested the average KL between the original and regularized weights across different models. All results are less than 0.1, indicating the effectiveness of this method.
>
> > Whether the direction magnitude decoupling idea transfers to encoder models such as BERT, Vision Transformer models, or multimodal models.
>
> This paper focuses on improving the effectiveness of VQ. **The core function of VQ is to reduce memory access**, which is generally more important for models with larger sizes. **The size of LLMs is usually much larger than that of BERT and ViT**, and previous VQ works have also been evaluated and experimented on LLMs.
>
> As for encoder models, we believe that PCDVQ is still effective. One of our core designs is the Standard Gaussian Regularization (see Section 3.2.1), which converts all weights into a uniform distribution ($N(0, 1)$), making the quantization algorithm **unaffected by the distribution characteristics** of model parameters.
>
> > Tasks that require stronger reasoning ability are needed.
>
> The five zero-shot QA tasks employed in our experiments are not merely simple fact-retrieval tasks but inherently **involve reasoning processes**.
>
> - WinoGrande [2] tests coreference resolution, a key component of logical reasoning that underpins more complex tasks like mathematical proof or code logic parsing.
>
> - Arc-Challenge [3] requires models to synthesize scientific knowledge and logical deduction to answer complex questions (e.g., inferring causal relationships between natural phenomena).
> - PIQA [4] demands physical commonsense reasoning to judge the feasibility of real-world actions (e.g., determining the correct sequence of steps to fix a broken object).
>
> Our results show that PCDVQ outperforms all SOTA baseline methods by at least 1.5% across six LLMs. This advantage in reasoning-involved QA tasks directly reflects PCDVQ’s effectiveness on tough tasks.
>
> **References:**
>
> [1]. QuIP: 2-Bit Quantization of Large Language Models With Guarantees.
>
> [2]. Winogrande: An adversarial winograd schema challenge at scale.
>
> [3]. Think you have solved question answering? try arc, the ai2 reasoning challenge.
>
> [4].  Piqa: Reasoning about physical commonsense in natural language.

---

> > ### Comment · Reviewer_3yCH · 2025-11-26
> >
> > Thank you for the authors’ response, which has resolved most of my concerns. I have also carefully reviewed the comments from the other reviewers. Since I am not an expert in this area, I maintain my original score.

---

> > > ### Author Response · Authors · 2025-11-26
> > >
> > > Thank you for your kindly feedback. We sincerely appreciate the time and effort you have dedicated to reviewing our work, and we are pleased that our responses have adequately addressed your concerns. Your valuable comments have significantly contributed to improving the quality of our manuscript.

---

### Official Review · Reviewer_ipFv · 2025-11-01

**Soundness:** 3
**Presentation:** 4
**Contribution:** 3
**Rating:** 6
**Confidence:** 2

**Summary:**

The paper introduces PCDVQ, a post-training weight-only quantization framework for LLMs that operates in polar coordinates: each weight vector is decomposed into direction and magnitude, which are quantized independently with a larger bit budget for direction. To reduce distortion, the method builds distribution-aligned codebooks. Across multiple LLMs and benchmarks in low-bit settings, PCDVQ consistently outperforms strong VQ and SQ baselines, indicating that decoupling and aligning to componentwise distributions yields better accuracy at very low precision.

**Strengths:**

1.The paper shows direction is markedly more sensitive to quantization than magnitude, and analyzes why Euclidean MSE emphasizes magnitude errors more strongly, supporting the decoupling design. The motivation of the work is clear and reasonable.

2.This work provides a clear and comprehensive theoretical foundation for polar coordinate decoupling, demonstrating strong depth and theoretical rigor.

3.Across multiple LLM families and standard zero-shot benchmarks, the main results tables show that PCDVQ generally matches or surpasses strong low-bit VQ/SQ baselines.

**Weaknesses:**

1.The choice to allocate more bits to direction is well supported by experiments, but the paper offers no formal analysis to guide the split or to select an optimal allocation under different conditions.

2. The method adopts a fixed vector dimension and borrows several settings from prior work, but it remains unclear how to adapt the dimension or the direction–magnitude bit split across model sizes, layer types, or differing weight statistics. Robustness to these design choices is not systematically examined.

**Questions:**

1.How sensitive is PCDVQ to design choices such as the direction similarity metric, codebook size, and the vector dimension, and are there general guidelines for setting these across models?

---

> ### Author Response · Authors · 2025-11-18
>
> We sincerely thank you for your valuable time and efforts in reviewing our manuscript. We have addressed each comment and made the necessary revisions to improve the quality and clarity of our manuscript.
>
> > Analysis to guide the split or to select an optimal allocation for the direction and magnitude.
>
> **Under a constrained total bit budget** (where 2-bit weight quantization corresponds to 16 bits usable for each vector), we observe a notable performance improvement when increasing the bit allocation for magnitude ($b$) from 1 to 2 bits. However, **further increasing consistently lead to performance degradation**.
>
> | Magnitude bits ($b$) of LLaMA-2-7B |   1   |     2     |   3   |   4   |   5   |
> | ---------------------------------- | :---: | :-------: | :---: | :---: | :---: |
> | average zero-shot accuracy (%)     | 49.75 | **58.60** | 52.32 | 44.05 | 34.98 |
>
> This observation suggests fixing $b$ to 2 and allocating other bits to the direction.
>
> > It remains unclear how to adapt the dimension or the direction–magnitude bit split across model sizes, layer types, or differing weight statistics.
>
> We propose a regularization technique in Section 3.2.1, which combines the Random Hadamard Matrix (RHM) and the standard Gaussian transformation. It transforms all weights **across different models** into the same distribution ($N(0, 1)$), making the quantization methods unaware to the original distribution.
>
> > Are there general guidelines for these settings (direction similarity metric, codebook size, and the vector dimension) across models?
>
> - For the codebook size, it is effected by the quantization bit-width and the vector dimension.
> - For the quantization bit-width, we set it to 2 and 2.25 for the comparison with previous 2-bit level baselines.
> - For the vector dimension, we set it to 8 as most works. Higher vector dimensions can model longer sequences, but leading to greater memory access overhead. We adopt a 8-dimension codebook because it can be fully accommodated within GPU L1 caches while providing sufficient accuracy to achieve state-of-the-art (SOTA) performance. **Further increasing vector dimensions may enhance model quantization performance but would degrade inference speed.**
> - For the direction similarity metric, we use the most common cosine similarity metric. A core issue raised in this paper is that the Euclidean distance tends to reduce the magnitude quantization error. This problem has been solved by the method of polar coordinate decoupling. As for the respective similarity matching algorithms for direction and magnitude, they are not the focus of this paper. We plan to advance research on this issue in subsequent studies based on the present framework.

---

> > ### Comment · Reviewer_ipFv · 2025-11-24
> >
> > I appreciate the authors' rebuttal and have decided to maintain my rating.

---

> > > ### Author Response · Authors · 2025-11-26
> > >
> > > Thank you for your reply. We have carefully addressed all the concerns raised in your review with detailed explanations in our response. Should any points require further clarification, please feel free to let us know. We look forward to the possibility of further productive discussions.

---

### Official Review · Reviewer_eGVU · 2025-11-01

**Soundness:** 3
**Presentation:** 3
**Contribution:** 3
**Rating:** 6
**Confidence:** 3

**Summary:**

The authors propose to decouple magnitude and direction quantization when performing vector quantization in large language models (LLMs). The proposed pipeline transforms vectors of weight matrices into polar coordinates and quantizes the magnitude and direction separately, taking into account their distinct statistical distributions. The paper presents a persuasive comparison with other scalar-based and vector-based quantization methods across a range of model sizes.

**Strengths:**

- The paper provides an insightful observation about vector-based quantization: the difference in the approximation behavior of direction and magnitude.
- The idea of decoupling magnitude and direction components and handling their different distributions is creative and conceptually elegant.
- The proposed method achieves strong performance compared to established baselines.

**Weaknesses:**

- The experiments primarily focus on a range of LLaMA models, with only a single Mistral experiment included. Broader evaluation across different architectures would strengthen the paper.
- The comparison with scalar-based quantization methods is limited and could be expanded for a fairer assessment.
- While the idea is simple and well-motivated, its conceptual simplicity raises questions about whether it is substantial enough for a full-length scientific paper.

**Questions:**

-Is there a difference in inference speed between scalar-based and vector-based quantization methods? If so, wouldn’t it be fair to include that in the comparison?
- Throughout the paper, you mention quantizing model weights one-by-one. Did you mean layer-by-layer?
- Were the results in Table 1 and Table 2 obtained without fine-tuning?
- Why was QuaRot not included in the experimental comparison, despite being mentioned in the paper?

---

> ### Author Response · Authors · 2025-11-18
>
> We sincerely thank you for your valuable time and efforts in reviewing our manuscript. We have addressed each comment and made the necessary revisions to improve the quality and clarity of our manuscript.
>
> >  Broader evaluation across different architectures would strengthen the paper.
>
> We conduct further experiments on the **MOE-based** Mixtral - 8x7B [1]. Results in the following table demonstrate the generalizability of PCDVQ. We will append more baselines and other series of models like DeepSeek-MOE to our manuscript.
>
> | Quantization Method | bit   | WikiText2$\downarrow$ | C4$\downarrow$ | HellaSwag$\uparrow$ |
> | ------------------- | ----- | --------------------- | -------------- | ------------------- |
> | -                   | 16    | 3.84                  | 6.87           | 64.88               |
> | GPTQ                | 2.125 | 48.63                 | 36.46          | 37.79               |
> | QuIP#               | 2     | 5.02                  | 8.81           | 58.42               |
> | VPTQ                | 2.25  | 4.95                  | 8.24           | 59.28               |
> | PCDVQ(ours)         | 2.25  | **4.33**              | **7.97**       | **60.91**           |
>
> > The comparison with SQ methods could be limited and unfair.
>
> The baseline of our experiment **includes both SQ and VQ methods**, as shown in the table below.
>
> | Quantization Method | Type |
> | :-----------------: | :--: |
> |        GPTQ         |  SQ  |
> |        GPTVQ        |  VQ  |
> |       DB-LLM        |  SQ  |
> |        QuIP#        |  VQ  |
> |        AQLM         |  VQ  |
> |        VPTQ         |  VQ  |
>
> > While the idea is simple and well-motivated, is it substantial enough for a full-length scientific paper?
>
> Our contributions lie not only in proposing the idea of "polar coordinate decoupling" but also in conducting motivation explanation, further method design, and experimental verification based on this idea:
>
> - We provide a **theoretical explanation** (see Section 3.1) for the differences in quantization sensitivity between direction and magnitude, as well as the inapplicability of existing Euclidean distance metrics.
> - We propose **DACC** to establish direction and magnitude codebooks according to their respective distributions (see Section 3.2.3).
> - We conduction **sufficient experiments** to verify the PCDVQ and its each components (see Section 4).
>
> > Is there a difference in inference speed between SQ and VQ? If so, wouldn’t it be fair to include that in the comparison?
>
> We compare the average QA accuracy and the "tokens per second" on LLaMA-2-7B. Experimental results are as follows and have been added to our manuscript.
>
> | Methods | Type | bits  | QA avg $\uparrow$ | tokens/s $\uparrow$ |
> | ------- | :--: | :---: | :---------------: | :-----------------: |
> | -       |  -   |  16   |       62.24       |        33.1         |
> | GPTQ    |  SQ  | 2.125 |       39.16       |        14.95        |
> | QuIP#   |  VQ  |   2   |       58.23       |        93.9         |
> | AQLM    |  VQ  | 2.29  |       58.57       |        20.6         |
> | VPTQ    |  VQ  | 2.02  |       58.13       |        40.6         |
> | PCDVQ   |  VQ  |   2   |     **58.60**     |      **95.7**       |
>
> > You mention quantizing model weights one-by-one. Did you mean layer-by-layer?
>
> The quantization objection is the weight matrices of all linear layers in LLMs. Thereby, "one by one" is equal to "layer by layer".
>
> > Were the results in Table 1 and Table 2 obtained without fine-tuning?
>
> Results in Table 1 and Table 2 are obtained with fine-tuning, including our PCDVQ and other baseline methods.  Fine-tuning is a common technique in weight-only quantization. We further conduct ablation study on this technique in Section 4.3.
>
> > Why was QuaRot not included in the experimental comparison?
>
> - This paper focuses on improving the effectiveness of VQ, while QuaRot [2] belongs to the SQ method.
> - This paper mainly studies improving the weight-only quantization of LLMs at extremely low bit levels (2-bit level), whereas QuaRot is aimed at 4-bit weight-activation quantization.
> - The quantization design of QuaRot for weights directly adopts the processing method of QuIP#, which has been included in the comparison methods of this paper.
>
> **References:**
>
> [1]. Mixtral of Experts.
>
> [2]. QuaRot: Outlier-Free 4-Bit Inference in Rotated LLMs.

---

### Comment · Area_Chair_Yxa9 · 2025-11-24

Dear Reviewers,

**We kindly encourage you to review and respond to the authors’ rebuttals**. Your timely feedback is important for ensuring a fair and thorough review process. Thank you for your contributions to ICLR 2026.

AC

---

### Note · Authors · 2026-02-10

**Comment:**

The order of authors cannot be reconciled, so it was decided to withdraw the manuscript.

**Withdrawal Confirmation:**

I have read and agree with the venue's withdrawal policy on behalf of myself and my co-authors.

---

### Meta-Review · Area_Chair_8vZa · 2026-01-04

**Summary:**

The reviewers generally found the paper to be technically sound and well motivated, with a clear insight that vector direction is more sensitive to quantization error than magnitude and that existing Euclidean-distance-based vector quantization methods do not adequately reflect this property. Several reviewers appreciated the conceptual elegance of polar coordinate decoupling and acknowledged that the proposed method achieves strong empirical results at extremely low bitwidths, particularly around the 2-bit regime, across multiple large language models.

However, a number of substantive concerns were raised that influenced the overall evaluation. One recurring concern was the limited scope of experimental validation. While the paper reports consistent gains on standard zero-shot benchmarks, multiple reviewers noted the absence of evaluations on more demanding reasoning benchmarks such as MMLU and GSM8K, where small quantization errors often lead to severe performance degradation. This raised questions about whether the reported improvements generalize to harder reasoning tasks or remain confined to relatively coarse-grained evaluation settings.

Another major concern involved system-level efficiency and inference cost. Several reviewers pointed out that the proposed framework introduces additional computational steps, including polar coordinate transformation, separate codebook searches for direction and magnitude, and inverse transformations during dequantization. Although throughput improvements were reported, reviewers emphasized that latency, wall-clock breakdowns, and energy efficiency were not initially analyzed in sufficient detail, particularly for edge or low-power environments where compute cost may dominate memory bandwidth considerations.

Reviewers also questioned the robustness and generality of several design choices. These included how the bit allocation between direction and magnitude is determined, how sensitive the method is to vector dimension, codebook size, and similarity metrics, and whether the Gaussian regularization assumption used for distribution-aligned codebook construction holds uniformly across layers, architectures, and model families. Some reviewers requested more systematic sensitivity analyses or clearer guidelines to support the applicability of the method beyond the specific configurations evaluated.

Concerns about conceptual novelty were raised by at least one reviewer, who noted significant overlap with prior work such as QuIP#, including shared components like E8 lattice-based codebooks and fine-tuning strategies. While the polar reinterpretation was acknowledged, this reviewer questioned whether the contribution goes beyond a reformulation of existing techniques and whether the essential novelty was sufficiently distinguished and emphasized.

Finally, reviewers sought clarification on experimental consistency and reproducibility. Questions were raised about whether all baselines were evaluated under identical fine-tuning regimes, whether performance gains could be attributed to differences in fine-tuning rather than the core quantization mechanism, and whether code, pretrained codebooks, and fine-tuning scripts would be released to enable independent verification and adoption.

Overall, while reviewers largely agreed that the idea is promising and empirically effective in the targeted low-bit setting, the concerns regarding the limited breadth of experimental evaluation, the lack of detailed system-level latency analysis, the robustness of key design assumptions, the degree of conceptual distinction from closely related prior work, and the clarity around experimental consistency and reproducibility collectively played an important role in shaping the final assessment of the paper.

**Reviewer Concerns:**

This meta-reviewer believes that a substantial portion of the reviewers’ concerns were meaningfully addressed in the rebuttal. In particular, the authors provided detailed clarifications and additional evidence regarding the core motivation of the work, namely the necessity of polar coordinate decoupling to properly capture the differing quantization sensitivities of direction and magnitude. The rebuttal strengthened this point both theoretically, by clarifying why directional components are inherently more error-prone under vector quantization, and empirically, by supplying extensive ablation studies and additional experiments that isolate the contributions of each component of the proposed framework.

Several technical concerns raised by the reviewers were also addressed with concrete experimental evidence. Questions regarding inference efficiency and system-level performance were answered through explicit measurements of end-to-end throughput and latency breakdowns, demonstrating that the additional operations introduced by the method do not adversely impact practical inference performance and, in some cases, improve it due to reduced memory access. Concerns about experimental consistency were similarly clarified, with the authors explaining their fine-tuning protocols in detail, aligning them with those used in prior work, and further supporting their claims through ablation results that show the proposed method remains effective even without fine-tuning.

The rebuttal was also responsive to questions about design choices and robustness. The authors provided empirical justification for the direction–magnitude bit allocation, discussed the role of Gaussian regularization in enabling global codebooks, and clarified how key hyperparameters such as vector dimension and similarity metrics were selected. These explanations helped reduce ambiguity around the method’s practical applicability and design rationale.

At the same time, some concerns remain partially outstanding. While the authors argued that their chosen zero-shot benchmarks capture essential aspects of reasoning, explicit evaluations on widely recognized hard reasoning benchmarks such as MMLU and GSM8K were not provided and are deferred to future work. Similarly, although the conceptual distinction from closely related methods such as QuIP# was clarified more clearly in the rebuttal, questions about the broader generality of the approach beyond extremely low-bit, weight-only quantization settings, as well as its behavior on non-decoder or non-language models, remain open.

Overall, the rebuttal was thorough and carefully executed, and it substantially improved the clarity, credibility, and technical depth of the submission. While some limitations in experimental scope persist, this meta-reviewer finds that the authors convincingly articulated why polar coordinate decoupling is a necessary and instructive perspective for low-bit vector quantization, and that the rebuttal demonstrated a high level of technical care, transparency, and engagement with the reviewers’ feedback.

**Reviewer Scores:**

Based on the content and tone of the rebuttal, this meta-reviewer believes that the reviewers who originally assigned higher scores would have viewed the authors’ responses positively had they been able to participate fully in the discussion. In particular, one reviewer who gave a score of 4 explicitly indicated in their comments that the clarifications and additional experiments addressed their main concerns and that the rebuttal strengthened their confidence in the work.

For the other two reviewers who also assigned scores of 4, while no explicit statement about score changes was provided, this meta-reviewer believes that they would have been reasonably satisfied with the rebuttal. The authors responded in detail to questions regarding system-level efficiency, design choices, and experimental consistency, and they supported their claims with additional measurements and ablation studies. These responses directly targeted the core issues raised in those reviews and demonstrated careful engagement with the feedback.

Overall, although it is difficult to assert that all reviewers would have formally increased their scores, this meta-reviewer believes that the rebuttal would have reinforced the positive assessments of the higher-scoring reviewers and alleviated several of their remaining concerns, leading to greater confidence in the technical soundness and contribution of the paper.

---

### Decision · Program_Chairs · 2026-01-26

Accept (Poster)